# Pathophysiology of Arginases in Cancer and Efforts in Their Pharmacological Inhibition

**DOI:** 10.3390/ijms25189782

**Published:** 2024-09-10

**Authors:** Patrycja Marzęta-Assas, Damian Jacenik, Zbigniew Zasłona

**Affiliations:** 1Molecure S.A., 101 Żwirki i Wigury St., 02-089 Warsaw, Poland; p.marzeta@molecure.com (P.M.-A.); damian.jacenik@biol.uni.lodz.pl (D.J.); 2Department of Cytobiochemistry, Faculty of Biology and Environmental Protection, University of Lodz, 141/143 Pomorska St., 90-236 Lodz, Poland

**Keywords:** arginase, cancer, metabolism, drug therapy, L-arginine, inhibitors

## Abstract

Arginases are key enzymes that hydrolyze L-arginine to urea and L-ornithine in the urea cycle. The two arginase isoforms, arginase 1 (ARG1) and arginase 2 (ARG2), regulate the proliferation of cancer cells, migration, and apoptosis; affect immunosuppression; and promote the synthesis of polyamines, leading to the development of cancer. Arginases also compete with nitric oxide synthase (NOS) for L-arginine, and their participation has also been confirmed in cardiovascular diseases, stroke, and inflammation. Due to the fact that arginases play a crucial role in the development of various types of diseases, finding an appropriate candidate to inhibit the activity of these enzymes would be beneficial for the therapy of many human diseases. In this review, based on numerous experimental, preclinical, and clinical studies, we provide a comprehensive overview of the biological and physiological functions of ARG1 and ARG2, their molecular mechanisms of action, and affected metabolic pathways. We summarize the recent clinical trials’ advances in targeting arginases and describe potential future drugs.

## 1. Introduction

Arginases (known as amidinohydrolases, EC 3.5.3.1) are binuclear metalloenzymes that were discovered in 1904 by Kossel and Dakin in the mammalian liver [1,2,3]. Arginases participate in many key enzymatic reactions, including the urea cycle, where they catalyze the conversion of L-arginine to L-ornithine and urea [1,2,3,4]. So far, two isoforms of arginases have been identified in mammals, i.e., arginase 1 (ARG1) and arginase 2 (ARG2) [1,2,5]. Although these two isozymes have a homologous enzymatic function, their tissue and subcellular distribution, genes, and chromosomal locations are different [1,6,7]. ARG1 is a much more explored isoform of arginase localized in the cytosol and is expressed mainly in the liver. ARG1 consists of 322 amino acids and is placed in the chromosome 6q23 region. ARG2, also called mitochondrial arginase, is expressed in extrahepatic tissues, i.e., the kidneys, small intestine, prostate, and mammary gland. ARG2 consists of 354 amino acids and is positioned in the chromosome 14q24 region [1,2,8,9].

ARG1, known as hepatic arginase, plays a major role in the urea cycle by removing toxic ammonia, while ARG2, in addition to regulating arginine homeostasis, is associated with the biosynthesis of polyamines, proline, citrulline, and glutamate, as well as nitric oxide (NO) (Table 1) [6,10,11]. Polyamines as organic compounds, together with proline, are necessary for controlling cell differentiation, proliferation, and viability, collagen synthesis, and tissue repair, as well as for regulating the immune response and immune system function [12,13,14,15,16]. The tumor microenvironment (TME) alters the immune response and modifies immune cell composition, resulting in immunosuppression [17,18]. The induced metabolism of arginases weakens the immune anti-tumor response by L-arginine depletion. In fact, arginine is the most depleted amino acid in the TME [17,19]. Myeloid-derived suppressor cells (MDSCs), tumor-associated macrophages (TAMs), and neutrophils are the main sources of ARG1 in the TME [17,18]. By contrast, ARG2 is mainly expressed in T regulatory (T_reg_) cells, where signaling through the mammalian target of the rapamycin (mTOR) pathway can improve their metabolic efficiency and regulate inflammation [17,20,21]. ARG2, depending on the access of L-arginine, is also able to regulate CD8^+^ T cells, influencing their functions of memory formation for self-renewal and transformation into effector T cells and regulating their anti-cancer function [17,21].

In recent years, both ARG1 and ARG2 have been linked to many diseases, such as inflammatory, cardiovascular, and neurodegenerative diseases, as well as various cancers, as described in detail in this review [22,23,24,25,26,27,28,29,30,31,32]. The accumulating evidence regarding the impact of arginases in many pathophysiological stages is linked with a growing interest in these enzymes in the area of drug discovery. Moreover, arginases are biomarkers and can be used to track disease progression [33]. Therefore, therapy targeting ARG1 and ARG2 has been proposed for various diseases. Currently, a dual inhibitor of arginases is under validation in clinical trials in patients with solid tumors; OATD-02 targets both extracellular and intracellular ARG1 and ARG2 [17,34]. In this review, we elaborate on the main physiological functions of ARG1 and ARG2, their metabolic pathways, and molecular mechanisms of action [1,35,36,37]. Additionally, based on recent experimental, preclinical, and clinical studies, we highlight how all these factors participate in the pathophysiology of many diseases. Lastly, we indicate the therapeutic candidates targeting arginases, their progress in clinical trials, and potential benefits for patients.

## 2. Arginases as Crucial Players in Physiology and Cancer Pathophysiology

### 2.1. Arginine Biosynthesis and Polyamine Production

Arginases, as enzymes of the urea cycle taking place in the liver, are primarily responsible for the catalysis of amino acids. In detail, ARG1 cleaves L-arginine into L-ornithine and urea, and this phenomenon allows for the removal of toxic ammonia in the last stage of the urea cycle [1,36,37,38]. However, in extrahepatic tissues, arginases are responsible for the hydrolysis of exo- and endogenous L-arginine [38]. L-ornithine obtained from the hydrolysis process can be metabolized in the decarboxylation reaction by ornithine decarboxylase (ODC, 4.1.1.17) to putrescine [37,38,39]. Putrescine is a substrate necessary for the synthesis of higher polyamines, such as spermine and spermidine [1,40,41]. Polyamines can also be produced from agmatine (endogenous guanido amine) through the activity of arginine decarboxylase (ADC, 4.1.1.4) [38]. In fact, polyamines are involved in the proper course of many metabolic processes; they are primarily responsible for cell growth and proliferation, as well as tissue repair [36,38,42]. As biogenic compounds, they stimulate the synthesis of phospholipids, ribonucleic acids (RNAs), deoxyribonucleic acids (DNAs), and proteins. Polyamines are components of cytoplasmic membranes and participate in the transport of metabolites and the regulation of ion channels [38,43]. Polyamines may also play an important role in carcinogenesis [38]. They are overexpressed in rapidly dividing cells such as cancer cells, e.g., in lung, breast, and colon cancer. This is related to the increased concentration of not only polyamines themselves but also genes and enzymes involved in their biosynthesis—including arginases [44,45,46]. López-Contreras et al. observed that polyamine levels were increased in non-small-cell lung cancer (NSCLC) with a change in arginine metabolism [47]. In the study conducted by Srivastava et al., stimulation of the ARG1 pathway was observed in head and neck squamous-cell carcinoma (HNSCC), as well as increased expression of ODC, which is essential for polyamine biosynthesis and promotes tumor growth [48]. The above studies indicate the pro-cancer characteristics of the increased activity of arginases and polyamine synthesis in various types of cancer [47,48,49].

### 2.2. Arginases and Proline and Glutamate Production

Further studies have indicated that arginases participate in the synthesis of proline and glutamate [1,38,50]. Mitochondrial arginase, i.e., ARG2, catalyzes reactions to produce L-ornithine, which in the mitochondria is transaminated to pyrroline-5-carboxylate with the participation of ornithine aminotransferase (OAT, 2.6.1.13) [38,50]. Proline is an important amino acid in collagen and is formed from pyrroline-5-carboxylate in the cytosol [38,50], playing an active role in shaping the TME [1,41]. Another amino acid produced from pyrroline-5-carboxylate is glutamate. Glutamate is a neurotransmitter and participates in many biochemical transformations. Its amide, i.e., glutamine, plays an important role in the body’s nitrogen metabolism [38,51]. L-ornithine, necessary for the synthesis of polyamines, is also a precursor for the synthesis of proline and glutamate. The inhibition of ARG2 activity would result in blocking the formation of L-ornithine from L-arginine in the arginase pathway and its transport from the cytosol to the mitochondrion—thus inhibiting the synthesis of both amino acids [38,50].

### 2.3. Arginases Control NO Production

Arginases regulate NO levels since L-arginine is the substrate for both ARG1, and ARG2, and can thus inhibit NO—a well-known endothelium-derived relaxing factor. NO modulates vascular tone, blood pressure, and hemodynamics; therefore, the dysregulation of NO signaling pathways contributes to the development of cardiovascular diseases but is also very relevant in cancer [8,36]. Increased levels of one of the arginases may cause nitric oxide synthase (NOS, EC 1.14.13.39) uncoupling, and this phenomenon is known as superoxide generation in cells. This event is related to reduction in NO generation and the enhanced production of O_2_^−^ and ONOO^−^ [37,52,53,54,55,56]. Small amounts of L-arginine may also cause NOS uncoupling and the increased release of O_2_^−^ [38,52,53,54,55,56]. Such damage caused by reactive nitrogen and oxygen species (RNOS) may contribute to cancer progression, which highlights the importance of a balance between these two enzymes [57]. The competition between the two catalytic enzymes, NOS and ARG1, is driven by the same cellular location in the cytosol. By contrast, ARG2 has a different intracellular localization compared to ARG1 and is manifested by a lower affinity for L-arginine than NOS [38,58,59]—important characteristics discriminating between the functions of ARG2 and ARG1. Finally, L-arginine can also serve as the sole substrate for NOS, which metabolizes arginine to NO and L-citrulline [36,38,60,61].

There are three different isoforms of NOS: inducible (iNOS), endothelial (eNOS), and neuronal (nNOS) [36,38]. To achieve full NO synthase activity, many cofactors are needed, such as flavin adenine dinucleotide (FAD), flavin mononucleotide (FMN), reduced nicotinamide adenine dinucleotide phosphate (NADPH_2_), molecular oxygen, tetrahydrobiopterin (BH_4_), and heme. The constitutive isoforms of NOS, i.e., nNOS and eNOS, produce small and short-lived amounts of NO, while iNOS is activated by various pro-inflammatory factors like interleukin (IL)-1, IL-6, interferon γ (INF-γ), macrophage migration inhibitory factor (MIF), and tumor necrosis factor α (TNF-α), as well as bacterial lipopolysaccharide (LPS), all of which produce large and long-lasting amounts of NO [5,51,60,61,62,63,64,65,66]. NO participates in many processes such as inflammation, thrombosis, regulates blood pressure, and neurotransmission, among others [62,67,68,69]. Enhanced arginase expression leads to a dysregulated production of NO and cancer progression. High levels of ARG2 and reduced levels of NO and iNOS were observed in prostate cancer cells, compared to non-cancerous tissues [70]. Inhibited NO production enhances angiogenesis and metastasis. Low NO concentrations result in the prevention of apoptosis and, subsequently, stimulation of the progression of cancer cells. At the molecular level NO stimulates signaling pathways such as mTOR, protein kinase B (PKB know as v-akt murine thymoma viral oncogene homolog 1, AKT), and extracellular-signal-regulated kinases (ERKs), which are necessary for the survival of cancer cells [41]. Conversely, a high concentration of NO causes cellular cytotoxicity and regulates the expression of one of the main DNA repair enzymes—DNA-dependent protein kinase catalytic subunit (DNA-PKcs)—and the expression of the p53 gene, which is critical for the apoptosis process and induces oxidative stress [71].

## 3. Cellular Consequences of Induced Arginase Expression

High levels of polyamines are observed in tissues that are characterized by high cell division activity, reflected by their presence in rapidly dividing cancer cells [72]. The TME consists of stromal and immune cells dependent on arginases, namely MDSCs, T and B cells, TAMs, and tumor-associated fibroblasts (CAFs) [1,73]. The microenvironment is important for cancer development and progression by interacting with nearby cells through the lymphatic and circulatory systems [74,75]. The overexpression of ARG1 and ARG2 in cancer impairs T cell function involving L-arginine, which is necessary for T cell differentiation and proliferation [76]. Vonderhaar et al. showed that increased ARG2 activity in the renal cell carcinoma (RCC) cell line depletes L-arginine and reduces CD3ζ expression. Moreover, the RCC cell line expressing ARG2 can modulate the L-arginine pathway and influence the function of T cells [77]. ARG1 is able to activate immunosuppressive cells such as MDSCs, which have the ability to suppress T cell activity and cause the escape of cancer cells from the immune response [78]. In melanoma, the overexpression of ARG2 in Treg increases their accumulation in tissues and suppressive capacity for the inhibition of mTOR signaling by the modulation of the PI3K/AKT/mTOR signaling pathway [1,20,79]. Enhanced activation of this pathway causes disturbances in cell proliferation and viability, which contributes to the formation of cancer and its further metastases (Figure 1). The regulation of arginases is quite complex. Chen et al. showed that ARG2 activation in pulmonary artery smooth muscle cells (PASMCs) was activated by the PI3K-AKT pathway [80]. AKT activation is followed by the phosphorylation of proteins such as caspase 9, Bad, glycogen synthase kinase-3 (GSK-3), nuclear factor-κB (NF-κB), p21Cip1, and p27 Kip1, which act as regulators of differentiation, proliferation, and apoptosis, as well as cell migration [81,82]. Similar to the AKT signaling pathway, mTOR kinase, which is a threonine-serine protein kinase, seems to modulate cell proliferation, growth, movement, and autophagy [1,81]. Yang et al. observed that ARG2 mediates the activation of mTORC1-S6K1 by myo-sin-IB (Myo1b), leading to lysosomal positioning, spatial separation, and the subsequent overactivation of the mTORC1-S6K1 signal, linking this phenomenon with aging and cell apoptosis [83]. In the most malignant thyroid tumors, ARG2 expression was increased compared to healthy tissues, and silencing the *Arg2* gene decreased AKT expression, causing apoptosis and reducing the expression of cell proliferation [84]. The overexpression of ARG1 in neuroblastoma cells increases the phosphorylated levels of ERKs and AKT [1,85]. IL-6 and IL-8 stimulate the production and subsequent secretion of ARG1 from MDSCs by stimulating the PI3K-AKT pathway, causing the suppression of T cells, which contributes to tumor development (Table 1) [86]. STAT3 and MAPK can also be linked to arginase action and modulate tumor growth and progression [1,14,87]. STAT3 is involved in tumor proliferation, invasiveness, angiogenesis, and migration [88]. Phosphorylated STAT3 can bind to the human ARG1 promoter region in MDSCs and thus regulate the immunosuppression of these cells [34]. STAT3 inhibition also causes a reduction in ARG1 expression in patients with acute myeloid leukemia (AML), partially restoring T cell proliferation [89]. Moreover, ARG2 can promote cancer metastasis through the mitochondrial H2O2-STAT3 pathway in melanoma [90].

### 3.1. Apoptosis

Arginases can regulate the process of apoptosis in various cells [36,91]. Métayer et al. conducted a study in which they evaluated the mechanism of arginase-induced cell death in precursor B-cell lymphoblasts using the pre-B ALL 697 cell line. They observed that arginase induced the rapid apoptotic death of B-ALL leukemic cells through arginine depletion, without killing normal human lymphocytes. Additionally, they described that arginase induced the activation of caspases, leading to increased cleavage of the endogenous substrate poly(ADP-ribose) polymerase (PARP) and caspase-3; chromatin condensations and exposure to phosphatidylserine were observed, which indicated the process of apoptosis. The overexpression of BCL-2 inhibited arginase-induced cell death but did not affect arginase-induced cytostasis [92]. In vascular smooth muscle cells (VSMCs), ARG2 induces apoptosis through mitochondrial dysfunction by complex positive crosstalk among p66Shc, S6K1-JNK, ERKs, and p53 [14]. In cells defective in argininosuccinate synthase (ASS, EC 6.3.4.5) and ornithine transcarbamylase (OTC, EC 2.1.3.3), pegylated human recombinant arginase (rhARG) induced intracellular apoptosis by depleting L-arginine and generating reactive oxygen species (ROS) [93]. Additionally, ARG2 in intestinal endothelial cells (ECs) can alleviate extracellular apoptosis induced by inflammatory cytokines through the competitive depletion of L-arginine and, consequently, reduce NO production, where excess NO can also lead to apoptosis (Table 1) [94].

### 3.2. Cell Senescence

Cellular senescence is a state of permanent and irreversible cell cycle arrest that occurs in proliferating cells subjected to stress [36]. We can distinguish two types of senescence: accelerated and replicative [95]. In human umbilical vein endothelial cells (HUVECs), Zhu et al. observed that ARG1 promoted eNOS uncoupling and induced the expression of senescence markers p21Cip1 and p53-S15, as well as the number of senescence-associated beta-galactosidase (SA-β-gal)-positive cells without S6K1 activation. The above-mentioned data indicating the involvement of arginases in the process of cellular senescence may be related to ROS generation during the uncoupling of NOS (Table 1) [96]. Yepuri et al. revealed a mechanism of reciprocal positive regulation between ARG2 and S6K1 in endothelial inflammation and aging. In fact, silencing *Arg*2 in ECs suppressed senescence markers such as p53-S15, p21, and β-galactosidase, and it activated the expression of intercellular adhesion molecule-1 (ICAM1) and vascular adhesion molecule-1 (VCAM1) [13]. Xiong et al. demonstrated that ARG2, together with Myo1b, mediates the activation of mTORC1-S6K1, leading to lysosome redistribution and subsequent cell senescence in VSMCs [83]. Notably, in non-cancerous cells, ARG2 overexpression in keratinocytes increased β-Gal expression, which is associated with cellular senescence [97].

### 3.3. Autophagy

Autophagy is recognized as the controlled breakdown of chemical molecules and cell fragments to maintain intracellular homeostasis under conditions of stress [98]. Autophagy in ECs is important for maintaining their plasticity and redox homeostasis and for the regulation of NO production and angiogenesis [98]. ARG2, independently from L-arginine-urea hydrolase in ECs, can suppress cellular autophagy through the activation of mTORC2 and AKT-mTORC1-S6K1 signaling and can cause the inhibition of AMPK [12]. Zhang et al. observed in their study that rhARG in bladder cancer cells induced autophagy by activating the ROS-mediated AKT/mTOR signaling pathway [99]. In another study, Wang et al. reported that rhARG impairs macrophage immune functions, such as TNF-α and IL-6 production, phagocytosis, and the surface expression of MHC-II, which results in the suppression of immune functions in activated macrophages through an autophagy-mediated mechanism [100].

### 3.4. Vascularization

The arginase-driven conversion of L-arginine into L-ornithine creates polyamines, necessary for cancer proliferation [36], but also structural cells such as fibroblasts and endothelium. Induced ARG1 expression accelerated rat VSMC proliferation and neointima formation through increased polyamine synthesis [101], while polyamine overproduction in macrophages increased cardiac fibroblast (CF) proliferation and collagen production [102]. The increased production of ARG1 and ARG2 may also influence EC proliferation linked to wound healing and angiogenesis [103]. Hypoxia in human pulmonary microcirculation ECs through ARG2 overexpression accelerates the proliferation of these cells by the epidermal growth factor (EGF)–epidermal growth factor receptor (EGFR)–extracellular signal pathway [104]. L-arginine is also necessary for cell growth and for the proliferation of brain microvascular ECs [36]. Hepatic ischemia–reperfusion causes increased arginase activity, which depletes the amount of L-arginine, subsequently causing cell cycle arrest and, consequently, inhibition of the proliferation [105].

### 3.5. Immune Response and Inflammation

Arginases are also significantly involved in the inflammation process [36], caused by physical, chemical, or biological factors [106]. Chronic inflammation is involved in changes in blood vessels, leading to cardiovascular diseases such as myocardial infarction, atherosclerosis, or exacerbated heart failure (HF) [107]. As a result of this process, blood vessels dilate and become more permeable so that various plasma proteins can reach the affected tissue. When harmful factors stimulate the external barrier, dendritic cells (DCs), mast cells, and macrophages are activated, which phagocytize to the side of inflammation and then secrete inflammatory mediators [106,107]. ARG1 is considered an anti-inflammatory marker of the M2 phenotype of macrophages, suppressing their inflammatory response. ARG1 expression in macrophages can be downregulated by Fos-related antigen 1 or upregulated by lipoproteins, and, as shown, activated macrophages challenged with rhARG1 can inhibit their immune response [100,108]. Conversely, ARG2 is associated with the M1 phenotype of macrophages and therefore may have pro-inflammatory properties compared to ARG1 [36]. T cells are the major cell components of the adaptive immune system responsible for the cellular immune response. Their proliferation is impaired in the presence of low L-arginine concentrations, so the depletion of this amino acid is associated with impaired immune response to tumor-associated neoantigens. Gołąb et al. described that ovarian cancer cells (OvCa) release ARG1 into the TME by extracellular vesicles (EVs) and then by DCs acquire suppressive properties and inhibit T-cell proliferation. Blocking arginase activity can restore appropriate L-arginine levels and T cell proliferative and anti-tumor potential [109].

ARG2 in senescent ECs triggers the release of inflammatory cytokines and chemokines such as IL-6 and IL-8 and the induction of ICAM-1 and VCAM-1 through the activation of p38MAPK and S6K1 (Table 1) [13,87]. In line with this, Huang et al. observed in their study that ARG2 induced age-related adipose tissue inflammation and inflammatory cell recruitment by secreting IL-6 and activation of p38 MAPK pathway [110]. Xiong and colleagues observed ARG2-induced TNF-α release in acinar cells mediated by p38 MAPK signaling, leading to β-cell apoptosis and subsequent decreased insulin secretion and glucose intolerance [111]. Liu et al. also observed the reduced secretion of pro-inflammatory cytokines such as IL-6 and TNF-α in *Arg2* knockout mice in bone marrow-derived macrophages, compared to WT macrophages [112]. Furthermore, Ming et al. found that the overexpression of ARG*2* in macrophages promotes their pro-inflammatory response through mitochondrial ROS generation, increasing the expression of monocyte chemoattractant protein-1 and the production of TNF-α, as well as IL6, contributing to atherogenesis and insulin resistance [16]. However, in myeloid cells, L-arginine depletion suppresses the immune response of T cells through the action of arginases [60]. T cells require an appropriate amount of extracellular arginases for proper proliferation and functioning. An L-arginine deficiency abolishes the formation of an immunological synapse, which inhibits T cell receptor (TCR) signaling and blockage of the cell cycle of T cells in the G0–G1 phase [60]. T cells are unable to up-regulate cyclin kinase 4 (CDK4) and cyclin D3, which is associated with reduced phosphorylation of the Rb protein and the low binding of transcription factor E2F1 [113].

Taken together, we can conclude that ARG1 and ARG2 play an often opposite role in the regulation of immune cell functions and the inflammatory response, which has consequences in the pathogenesis of metabolic and cardiovascular diseases.

## 4. Arginases Regulate Metabolic and Mitochondria Machinery

### 4.1. ARG2 Regulates Oxidative Metabolism

Mitochondria are responsible for producing adenosine triphosphate (ATP) through oxidative phosphorylation [114]. These organelles are involved in the production of ROS, the regulation of Ca^2+^ homeostasis, and amino acid metabolism. Mitochondria maintain homeostasis through their dynamic cycles like biogenesis, fusion, and fission, as well as mitophagy, as a response to unfolded proteins [114,115]. Due to the fact that ARG2 is located in mitochondria, it plays the main role in mitochondrial functions. Mitochondrial ARG2 present in ECs can be activated by various stimuli such as thrombin, TNFα, H_2_O_2_, the EGF, and LPSs, and its expression can be reduced by compounds such as simvastatin, cocoa flavanols, and genistein [116]. Pandey et al. showed that oxidized low-density lipoprotein (oxLDL) can induce the activation of ARG2, which is regulated by mitochondrial processing peptidase (MPP). MPP cleaves N-terminal cleavable presequences and plays a role in the transport of mitochondrial proteins; therefore, it is responsible for the reverse translocation from mitochondria to the cytoplasm. In mitochondria, ARG2 reduces L-arginine production, causing eNOS uncoupling [116,117,118,119,120,121]. Subsequently, reduced synthase activity leads to reduced NO production and the overproduction of ROS and superoxides, which in turn lead to oxidative stress, which continuously activates mitochondrial-to-cytoplasmic translocation by MPP and ARG2 activation (Table 1).

ROS also play a key role in regulating the oxidation of lipoproteins, including oxLDL, which in turn induce ARG2 by ROCK and MPP, and thus forms a feedback loop [116,117,118,119,120,121]. Hwang et al. described how eNOS can be regulated by various extracellular signals, e.g., by Ca^2+^, which in mitochondria regulates ATP synthesis and the rate of oxidative phosphorylation. It has been reported that ARG2 can regulate both mitochondrial and cytolytic Ca^2+^ concentration. ARG2, together with the p32 protein in endothelial mitochondria, played an important role in modulating intracellular NO, ROS, and Ca^2+^ concentrations [122]. Zhang et al. in their study specified that ARG2 present in hepatocytes induced mitochondrial ureahydrolysis and drove hepatic oxidative metabolism, leading to glucose oxidation in mammals [123]. ARG2 present in hepatocytes enhances peripheral and hepatic AKT phosphorylation, which increases peripheral energy expenditure and inhibits hepatic steatosis, regardless of the hydrolytic activity. Mitochondrial ARG2 is able to control glycemia in the obese state driving glucose oxidation. Lim et al., in their study, confirmed that in murine endothelium, ARG2 was mainly confined to mitochondria regulating NO production and vascular endothelial function by modulating eNOS activity [124]. Mitochondria are also key organelles for the polarization of macrophages. M1 macrophages, through the expression of NOS, are responsible for the synthesis of arginine into NO and citrulline. By contrast, M2 macrophages express more ARG1, which hydrolyzes arginine to urea and ornithine [125]. Uchida et al. used mice with acute kidney injury (AKI) induced by cisplatin to test whether ARG2 could be a therapeutic target for AKI. Cisplastin-treated *Arg2* knockout mice had significant improvement in renal dysfunction through a characteristic process of apoptosis and an apparent reduction in acute tubular injury compared to WT mice [126]. LPS administration increases the mitochondrial membrane potential, reducing ROS secretion dependent on ARG2 [126]. In summary, ARG2 expression correlates with the inflammatory response of macrophages through the secretion of ROS and, consequently, the exacerbation of AKI [126].

### 4.2. ARG2’s Specific Role in Mitochondrial Function

Accumulating evidence has highlighted the essential role of ARG2 in the regulation of cancer progression, suggesting that the inhibition of ARG2 may lead to clinical benefits for patients with cancer. The role of ARG2 in pancreatic cancer was suggested by Zaytouni et al., who observed that the silencing or loss of *Arg2* suppresses obesity-driven pancreatic cancer development [127]. Líndez et al. demonstrated how mitochondrial ARG2 determines the activity of CD8^+^ T cells. The adoptive transfer of CD8^+^ T cells with *Arg2* knockout reduced tumor growth in murine model of melanoma and colon cancer. The anti-cancer function of CD8^+^ T cells with *Arg2* knockout is related to enhanced cytotoxicity, memory formation, and the persistence of anti-cancer CD8^+^ T cell responses [21]. The above-mentioned events are independent from ARG1 presence. The significance of ARG2 is fundamental not only for cancers but also for inflammation, especially for macrophage function. Hardbower et al. reported that the knockout of *Arg2* in *Helicobacter pylori*-infected mice led to increased regulation of polyamine metabolism and the activation of M1 macrophages, which is associated with increased gastritis. ARG2 may play an inflammasome function in macrophages, as pro-IL-1β cleavage into mature IL-1β [128]. Data obtained by Dowling et al. documented that ARG*2* expression is mediated by IL-10/microRNA-155 axis in inflammatory macrophages, and this phenomenon affects the dynamics and oxidative respiration of mitochondria and determines the inflammatory stage of macrophages [129]. The functions of arginases in metabolic and mitochondrial mechanisms indicate that both enzymes play key roles in proper cellular functioning and are involved in the pathophysiology of many diseases; therefore, it is important to maintain an appropriate balance, especially between arginases and NO, which compete together for arginine (Figure 2).

## 5. Significance of Arginases in Non-Cancer Diseases

The function of arginases is well documented in human pathophysiology, and their disturbed expression and action are observed in many diseases. Numerous studies have observed the link between arginases, NOS, and NO in cardiovascular diseases [37]. Decreased NO production contributes to the increased adhesion potential of leukocytes to the endothelium, impaired vasorelaxation, and the deposition of L-ornithine, which is related to fibrosis and the stiffening of vessels [37,130]. In the case of hypertension, increased arginase activity, decreased amounts of NO, L-arginine, and BH_4_, and increased O_2_^−^ production cause impaired endothelium-dependent vascular relaxation [131,132]. Increased ARG2 activity has also been observed in pulmonary arterial hypertension (PAH). The increased protein level of ARG2 attenuates the endothelium-dependent dilation of pulmonary segments in the presented pulmonary embolism model [133,134].

Increased ARG1 and a decrease level of NO is observed in sickle cell disease (SCD), where during hemolysis, red blood cells (RBCs) release large amounts of ARG1, which contributes to reduction in NO and L-arginine and increased levels of L-ornithine [135,136,137]. The activation of ECs, reduced amount of NO, and the deposition of sickled RBCs in capillaries causes the pathology to deepen.

It has been shown that arginases may also be key factors causing cardiovascular problems in diabetic patients [37]. Chandra et al. presented data showing that diabetes or hyperglycemia may increase the level of arginases in aortic ECs through the Ras homolog family member A (RhoA)/ Rho-associated protein kinase (ROCK)/MAPK pathway [138]. The activation of ARG1 in RBCs in patients with type 2 diabetes causes dysfunction in the endothelium of healthy arteries through a ROS-dependent mechanism [139].

Most studies have linked the increased activation of arginases with neurological diseases such as AD, multiple sclerosis (MS), Parkinson’s disease (PD), stroke, and traumatic brain injury [37,140]. Both ARG1 and ARG2 are expressed in neurons and alter NO signaling, which is the main cause of neurological disorders and brain damage [2]. In the case of long-term cerebral ischemia, NO is synthesized with the contribution of iNOS, which is activated by released cytokines [38]. NO, which enters synapses, is responsible for the increased secretion of glutamate and Ca^2+^, which then leads to the promotion of apoptosis or necrosis of nerve cells [38,141].

Zimmerman et al. noted a particular increase of the *Arg1* gene expression in lung tissues of mice with an experimental asthma model [142]. Nevertheless, Ceylan et al. observed that the L-arginine/NO pathway is involved in the pathophysiology of asthma [143]. Pawliczak and Lewandowicz reported that both IL-4 and IL-13 strongly induced arginase expression in the lungs of patients with asthma [144]. Arginases as precursors of polyamines and proline modulate the activity of NO synthase, and this event is involved in the remodeling of the airways in inflammation-induced asthma [38,145]. Additionally, it is worth noting that phenotypes of mice with liver-specific KO of *Arg1* and phenotypes of mice with total-body KO of *Arg1* are similar to each other and in both cases are lethal [146,147]. The increased activity of ARG1 and its enhanced expression is associated with the release of cytokines, catecholamines, and the transforming growth factor β (TGF-β), and this results in the down-regulation of citrulline, arginine, and NO production [38,148]. Increased levels of arginases and L-ornithine, which are necessary for the synthesis of polyamines and proline, may play a significant role in the wound-healing process, so the overexpression of arginases can also be observed in injuries, exactly in macrophages, which occur in increased amounts in the area of damaged tissues [1,38,51,148].

## 6. Clinical Perspectives of Arginase Inhibitors

### 6.1. Arginase Vaccines and Pegylated rhARG in Clinical Trials

For some time now, scientists have begun to develop immunomodulatory vaccines targeting arginases to improve the anti-cancer response in cancer immunotherapy [1]. Andersen and other colleagues conducted preclinical trials where immunogenic peptides derived from the ARG1 protein sequence were used and noted that these peptides are able to activate T cell-specific ARG1 and suppress ARG1-expressing malignant cells [149,150,151]. In phase I of a clinical trial (NCT03689192), a vaccine containing ARG1 peptides was tested in combination with immune checkpoint inhibitors, i.e., anti-programmed death receptor 1 (PD-1). Vaccines were administered subcutaneously to patients every third week. The study used enzyme-linked immunosorbent assays and intracellular cytokine staining to assess the vaccine-specific immune response. It was documented that co-administration of ARG1 vaccine with anti-PD-1 antibodies created a pro-inflammatory microenvironment and changed the M1/M2 macrophage ratio in the tumor, reducing tumor suppressive immunity [152,153]. Adverse events reported in the study included injection site reactions and shoulder arthralgia [152,153]. There was a risk of inducing autoimmune reactions in patients due to the fact that ARG1 is expressed in the liver and other non-cancerous tissues. However, increased transaminase levels were noted only in patients with progressing metastatic lesions in the liver [152,153]. The same group developed a study using 34 peptides covering the entire ARG2 sequence and assessed the reactivity of these peptides in peripheral blood mononuclear cells (PBMCs) obtained from healthy donors and cancer patients. In a Lewis lung cancer tumor model, they found that vaccines with these peptides significantly inhibited tumor growth by activating ARG2-specific T cell response, which may constitute a future potential for a new therapeutic candidate in immunotherapy [154]. Chan et al. showed that the growth of hepatocellular carcinoma (HCC) depends on exogenous arginine, which cannot be hydrolyzed due to the weak expression of ASS and OTC in these cells. This group in phase II of the clinical trial (NCT01092091) verified pegylated rhARG1 (PEG-BCT-100) as an arginine-depleting drug, which has potential in the treatment of advanced HCC [155]. PEG-BCT-100 was administered intravenously to patients at a dose of 2.7 mg/kg weekly [155]. Adverse events after the administration of PEG-BCT-100 reported by patients included limb edema, fatigue, constipation, and anemia [155]. Chiang et al. also conducted a study (NCT01551628) in patients with relapsed or refractory leukemia or lymphoma using rhARG1 to show that the drug was safe and effective, but the final report from the study is currently unavailable. Zhang et al. conducted a study on bladder cancer cell lines, establishing that rhARG1 (BCT-100) induced autophagy and apoptosis through the ROS-activated AKT/mTOR signaling pathway, exhibiting anticancer effects [99]. The use of rhARG1 may prove to be a therapy for some types of cancer in the future.

### 6.2. Naturally Occurring Arginase Inhibitors

Inhibitors of arginases can be divided into two classes, naturally occurring compounds and synthetic compounds. Naturally occurring arginase inhibitors include chlorogenic acid (CA) and piceatannol (PIC), and synthetic compounds include N(ω)-hydroxy-l-arginine (NOHA) and its analogs, such as S-(2-boronoethyl)-L-cysteine (BEC), 2(S)-amino-6-boronohexanoic acid (ABH), L-norvaline, and CB-1158 [1]. PIC is a polyphenol and belongs to the natural inhibitors of arginases [156]. Its effect on mammalian ARG1 was determined to be an IC_50_ of 12.1 µM [157]. Song et al. reported that PIC, through pathways related to p38, p-STAT3, NF-ĸB, HIF-1α, and c-Myc, induced apoptosis and limited the growth and metastasis of breast cancer [158]. From the cellular point of view, PIC induces apoptosis and arrests the cell cycle in the G0/G1 phase by inhibiting cyclin-dependent kinase (CDK) activity. This event led to reduction in bladder cancer cell proliferation [159]. Its anti-cancer properties have also been reported in lung cancer, apart from and in combination with gemcitabine [160,161]. CA inhibited ARG1 in mammals with IC_50_ of 10.6 μM [157]. CA is able to induce apoptosis, and its anti-cancer function is linked to antioxidant properties [162,163,164]. These compounds, however, show low potential for arginase inhibition and a lack of specificity. Nevertheless, the CA structure may be a promising starting point for creating new and more potent inhibitors of arginases.

### 6.3. Synthetic Arginase Inhibitors

Previously identified and developed inhibitors of arginases, such as norvaline as a substrate for aminotransferases, had low potency and, therefore, side effects due to the high concentrations required [165]. α-difluoromethylornithine (DFMO) is a potent irreversible inhibitor of ODC; however, no significant effect on arginase activity has been determined [166]. NOHA and nor-NOHA belong to the group of hydroxy arginine derivatives; both are competitive and reversible inhibitors of arginases. NOHA is produced during the breakdown of L-arginine into NO and L-citrulline using NOS. nor-NOHA is an inhibitor with a longer half-life than NOHA; the Kd of nor-NOHA is 0.517 µM, while for NOHA it is 3.6 µM for human ARG1. The Ki for nor-NOHA against ARG2 is 51 nM, and for NOHA the Ki is 1.6 μM [167,168]. The inhibition of arginases by NOHA has pro-apoptotic and anti-proliferative effects in colorectal and breast cancer cells [169,170]. Zhu et al. showed that the use of nor-NOHA inhibited ARG1 in HepG2 cells, induced the apoptosis process, and inhibited the migratory and invasive abilities of these cells [171]. Ludwig et al. demonstrated that nor-NOHA, by reversing the tumor-growth-promoting effect of ARG1 by TAM-derived exosomes, may play a significant therapeutic role in the treatment of glioblastoma [172]. Alonso et al., using murine ovarian cancer, reported that the application of nor-NOHA blocked ARG1 activity and reversed the immunosuppression of vascular leukocytes (VLCs) in ovarian cancer [79]. In line with this, Javrushyan and his colleagues reported that the administration of nor-NOHA to rats with a 7,12-dimethylbenzathracene-induced mammary tumor reduced the size and number of these tumors [49]. In phase II of clinical trials, nor-NOHA was tested in the treatment of cardiovascular diseases. The use of nor-NOHA improved the function of the microvascular endothelium in patients with type 2 diabetes mellitus (T2DM, NCT02687152) and in patients with coronary artery disease (CAD). Additionally, it protected against endothelial dysfunction after ischemia–reperfusion (NCT02009527, Table 2) [173,174]. Finally, nor-NOHA reduced the development of atherosclerosis and hypertension [175]; however, poor specificity for the isoforms of both enzymes still limits its use [1,168]. Currently, no clinical trials using NOHA or nor-NOHA are ongoing, and the last clinical trial data on these compounds were reported in 2023.

ABH and BEC are further boronic acid derivatives of arginase inhibitors that can bind to the active manganese site of arginases. Both inhibitors can competitively inhibit arginase activity. The IC_50_ of ABH is 1.54 µM for human ARG1 and 2.55 µM for human ARG2 [176]. BEC is a cysteine-based ABH analog, its Kd for human ARG1 is determined to be 270 nM, and the Kd for ARG2 is determined to be 220 nM [58,177,178]. It has been reported that the inhibition value of both these inhibitors is dependent on the pH level, and the affinity for both arginases is different [179]. The selectivity of ABH is lower towards ARG1 compared to ARG2 [180]. Abdelkawy et al. determined that both compounds are characterized by low bioavailability, and only 5% of BEC and ABH is detectable after single oral administration, while after intraperitoneal multiple-dose administration, bioavailability increased to 50% [181]. In Fischer rats, oral administration of ABH at a dose of 400 mg per day for 25 days did not lead to toxicity and side effects [182]. However, it was reported that compounds with a boronic acid head group achieved unacceptable toxicity to normal cells [182]. The above-mentioned data strongly suggest that further research is needed to determine the appropriate safe dose, as well as the structural optimization of ABH and BEC to make them promising candidates for clinical use in cancer therapy.

The arginase inhibitor, CB-280, was tested in phase I of clinical trial in patients with cystic fibrosis (CF) (NCT04279769, Table 2). In this study, the authors examined the safety, pharmacokinetics, pharmacodynamics, and biological activity of CB-280. They predicted that CB-280 will be able to inhibit arginases, thereby restoring endogenous NO production, leading to improved respiratory function, as well as antimicrobial response in CF patients, but the results are still not available [183]. Numidargistat (CB-1158) developed by Calithera Biosciences is a small molecule and ARG1 inhibitor, administrated orally, with an IC_50_ of 98 nM (Table 2) [184,185]. Steggerda et al. showed that CB-1158 inhibited ARG1, creating a pro-inflammatory environment, reducing tumor growth and reversing the immunosuppression of TME [33]. CB-1158, in combination with other chemotherapeutics and/or immunotherapies, may provide clinical benefits for patients. NCT03314935 and NCT02903914 were clinical trials where CB-1158 was investigated as a single agent, as well as in combination with immune checkpoint inhibitors, i.e., pembrolizumab or other chemotherapeutic agents, in the treatment of advanced and metastatic solid tumors [186,187,188].

Currently, in phase I of clinical trials is OATD-02, developed by Molecure S.A. (NCT05759923, Table 2). OATD-02 is a boronic acid derivative and a potent dual inhibitor of ARG1 and ARG2, with an IC_50_ of 20 nM and 48 nM, respectively (Figure 3) [189]. Additionally, this is the first study to confirm the benefits of ARG2 inhibition in cancer treatment. Błaszczyk et al. and Borek et al. have demonstrated the anti-cancer and immunomodulatory efficacy of OATD-02 in animal models, as well as its moderate bioavailability and long half-life after oral administration [17,189]. OATD-02 differs primarily from CB-1158 in potency, is characterized by better arginase selectivity, and showed greater anti-tumor activity in in vivo studies [17]. In a study using the syngeneic Renca model, OATD-02 reduced regulatory T cells (T_reg_) infiltration, thereby improving the CD8^+^/T_reg_ ratio [17], and inhibited the proliferation of K562 human leukemic cells [17]. The anti-cancer efficiency of OATD-02 has also been reported in xenografts induced by GL261 glioma, B16F10 melanoma, CT26 colorectal adenocarcinoma, Lewis lung cancer, and ID8 ovarian cell administration [17,189,190,191,192]. Błaszczyk et al. observed that OATD-02 reduced glioma growth by modifying the TME and increased the anti-cancer effect of anti-PD-1 antibodies, affecting the response of NK and myeloid cells [17,191]. Grzybowski et al. conducted a study using an OATD-02 inhibitor in combination therapy with a STING agonist (DMXAA) and immune checkpoint inhibitors, thereby improving the anti-tumor response [17,190]. OATD-02 may demonstrate effectiveness in the treatment of immune-related cancers and in cancers with enhanced ARG2 expression, such as pancreatic cancer, chronic and acute myeloid leukemia, kidney cancer, and in patients with prostate cancer [17,77,127,193,194,195]. It can therefore be concluded that OATD-02, as the dual arginase inhibitor, may improve anti-cancer therapy, which seems to be linked to the inhibition of cellular arginases, ARG2-dependent metabolic adaptations, and ARG1 located in EVs.

## 7. Conclusions

Arginases, as enzymes involved in the hydrolytic breakdown of arginine to ornithine and urea, serve important functions in cellular metabolism. The deregulation of ARG1 and ARG2 activity are key in the initiation and development of many human diseases. From a functional point of view, the competition of arginases with NOS for L-arginine, as well as their participation in proline synthesis, may play a crucial role in diseases such as injuries, inflammation, stroke, and cardiovascular diseases. Moreover, arginase-mediated synthesis of polyamines is directly associated with tumorigenesis and cancer development.

Arginases regulate cancer cell proliferation, apoptosis, and migration and affect immunosuppression involving not only L-arginine metabolism but also numerous other signaling pathways. Accumulating studies indicate that the finding of an appropriate candidate that inhibits the activity of ARG1 and ARG2 may be crucial in the therapy for many human diseases. Recently, numerous clinical trials have been developed addressing inhibitors of arginases for the treatment of cancer, such as CB-1158 and OATD-02. Based on the homology of the enzymatic active sites of human ARG1 and ARG2, which is quite high, finding a specific inhibitor targeting a specific isoform is challenging. Moreover, further clinical trials have to be conducted to help determine the effects of the long-term use of inhibitors of arginases in patients.

## Figures and Tables

**Figure 1 ijms-25-09782-f001:**
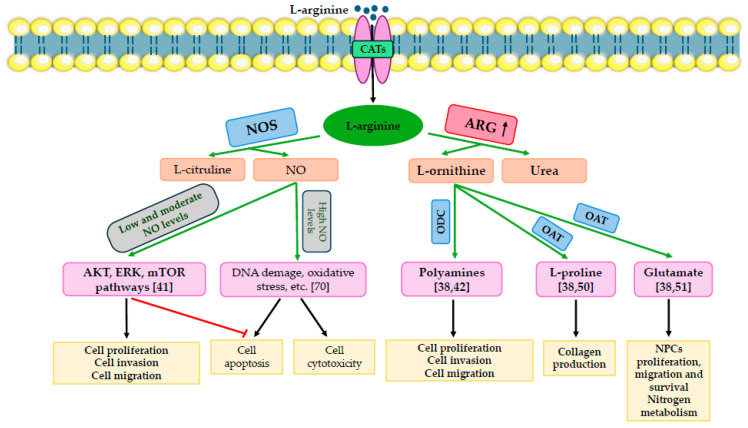
Cellular and molecular consequences of the imbalance of NO and L-arginine action in cancer. NOS metabolizes L-arginine to generate both L-citrulline and NO. At high NO concentrations, cellular apoptosis, oxidative stress, DNA damage, and cellular toxicity occur. Moderate and low amounts of NO prevent cells from apoptosis through the ERK, mTOR, and AT signaling pathways, and they promote the development of cancers and the invasion of cancer cells. Arginases hydrolyze L-arginine to generate L-ornithine and urea. In the reaction catalyzed by OAT, L-ornithine is transformed to L-proline and glutamate. ODC catalyzes the reaction of polyamine generation from L-ornithine. L-proline participates in the production of collagen. In the case of cancer, the balance between NO and L-arginine is disturbed. There is increased arginase activity, which depletes L-arginine to a greater extent, causing reduced NO production and thus increasing L-ornithine levels. Then, L-ornithine catalyzed by OAT and ODC increases the proline, glutamate, and, most importantly, polyamine generation. In this way, cancer cell migration, invasion, and proliferation occur. Because the NO level is low, there is no cellular apoptosis by the AKT, ERK, and mTOR pathways but further proliferation of these cells. OAT, ornithine aminotransferase; NOS, nitric oxide synthase; ARG, arginase; ODC, ornithine decarboxylase; NO, nitric oxide; AKT, protein kinase B; CAT, cationic amino acid transporter; mTOR, mammalian target of rapamycin; ERK, extracellular signal-regulated kinase; NPCs, neural progenitor cells.

**Figure 2 ijms-25-09782-f002:**
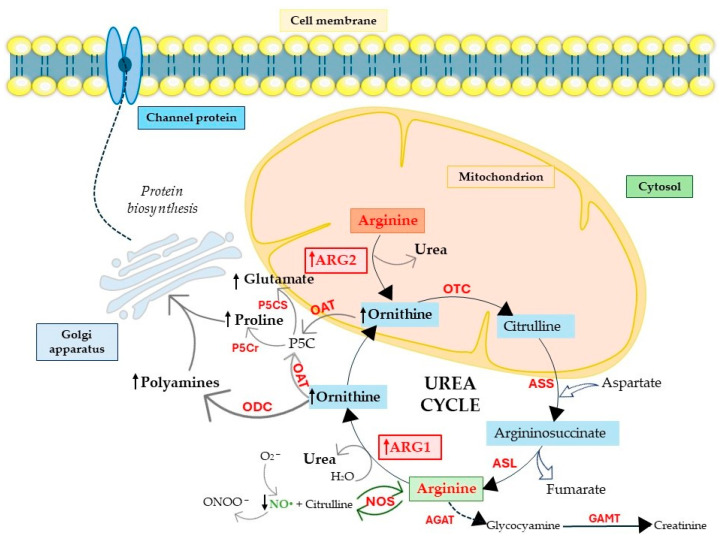
The role of arginase in cancer metabolism. In the urea cycle, the carbonyl group of citrulline, together with the amino group of aspartate, forms argininosuccinate in a condensation reaction. This reaction is catalyzed by ASS and is ATP-dependent. Argininosuccinate is then cleaved by ASL to form fumarate and arginine. Arginine and glycine in the urea cycle are catalyzed by glycine amidinotransferase AGAT (EC 2.1.4.1) to produce glycocyamine, which is further methylated to creatinine by guanidinoacetate methyltransferase GAMT (EC 2.1.1.2). Arginine in the urea cycle is also cleaved by ARG to ornithine and urea, and by NOS to NO and citrulline. Ornithine is necessary for the further synthesis of polyamines, proline, and glutamate by ODC and OAT synthases, respectively. Ornithine is further transported back to the mitochondria where it is converted to citrulline by OTC to begin the urea cycle again. In the case of cancer, ARG1 and ARG2 become activated, so arginine is fully depleted. Arginine is not hydrolyzed by NOS, and, as a result, the amount of NO is reduced, which disturbs the synthesis of chemokines and pro-inflammatory cytokines responsible for the activation of T cells. Activated arginases lead to the increased production of ornithine, which is a direct substrate for the synthesis of polyamines, glutamate, and proline. This effect leads to increased protein biosynthesis and the proliferation and migration of cancer cells. ARG, arginase; ODC, ornithine decarboxylase; NOS, nitric oxide synthase; ASL (EC 4.3.2.1), argininosuccinate lyase; ASS, argininosuccinate synthase; OTC, ornithine transcarbamylase; OAT, ornithine aminotransferase; P5Cr (EC 1.5.1.2), P5C reductase; P5CS (EC 2.7.2.11), P5C synthase; P5C, Δ1-pyrroline-5-carboxylate; AGAT, glycine amidinotransferase; GAMT, guanidinoacetate methyltransferase, → direct path; −→ multistep path.

**Figure 3 ijms-25-09782-f003:**
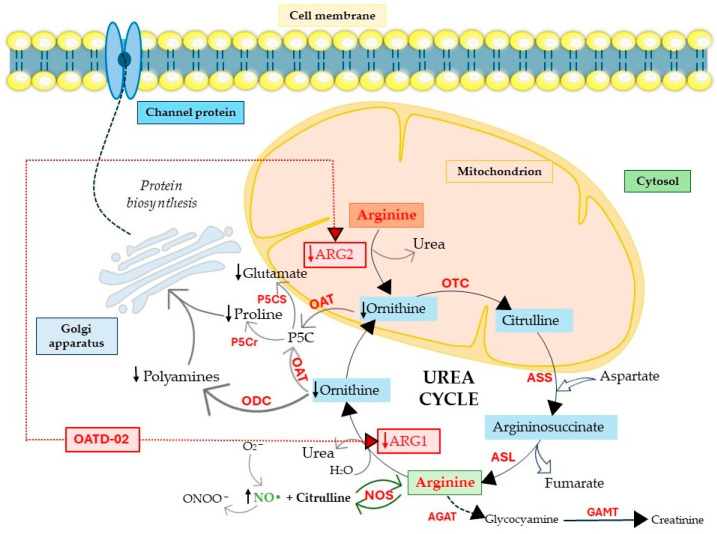
OATD-02 inhibits intracellular ARG1 and ARG2. The ARG1/ARG2 inhibitor OATD-02 inhibits the activity of cytosolic ARG1 and mitochondrial ARG2 by preventing the degradation of arginine by arginase. Arginine can therefore use NOS to stimulate NO synthesis and the production of chemokines and pro-inflammatory cytokines, which leads to T cell activation and proliferation. The inhibition of both arginases causes reduced production of L-ornithine. ODC does not catalyze the decarboxylation of L-ornithine to form putrescine, a substrate necessary for the synthesis of polyamines, which translates into reduced polyamine generation, thus inhibiting further proliferation of cancer cells. Additionally, OAT does not catalyze the reaction of converting L-ornithine to 5PC, which translates into reduced proline and glutamate generation. ARG, arginase; ODC, ornithine decarboxylase; NOS, nitric oxide synthase; ASL, argininosuccinate lyase; ASS, argininosuccinate synthase; OTC, ornithine transcarbamylase; OAT, ornithine aminotransferase; P5Cr, P5C reductase; P5CS, P5C synthase; P5C, Δ1-pyrroline-5-carboxylate; AGAT, glycine amidinotransferase; GAMT, guanidinoacetate methyltransferase, → direct path; −→ multistep path.

**Table 1 ijms-25-09782-t001:** ARG1 and ARG2 regulate essential processes involving several mediators.

Arginase Isoform	Substrate	Mediator	Process
ARG1	L-arginine	NO	Apoptosis
		ROS	Cell senescence, apoptosis, inflammation
		IL-6 IL-8	Suppression of T cells
ARG2		NO	Apoptosis
		ROS	Cell senescence, apoptosis, inflammation
		IL-6 IL-8 TNF-α	Inflammation

**Table 2 ijms-25-09782-t002:** Inhibitors of arginases in clinical trials.

Compound	Clinical Trial ID	Phase of Clinical Trial	Condition or Disease
N(ω)-hydroxy-nor-L-arginine (nor-NOHA) 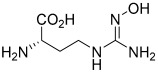	NCT02687152	Phase I (completed)	Type 2 Diabetes Mellitus
N(ω)-hydroxy-nor-L-arginine (nor-NOHA) 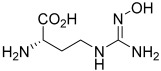	NCT02009527	Phase I (completed)	Ischemia–Reperfusion Injury
Numidargistat (CB-1158) 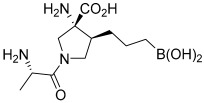	INCB001158	Phase II (completed)	Metastatic Cancer; Colorectal Cancer; Lung Cancer; Solid Tumors; Gastric Cancer; Head and Neck Cancer; Bladder Cancer; Mesothelioma; Renal Cell Carcinoma; Urothelial Cancer
CB-280 (chemical structure of the compound undisclosed)	NCT04279769	Phase I (completed)	Cystic Fibrosis
OATD-02 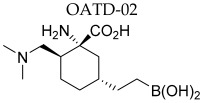	NCT05759923	Phase I (active)	Advanced/Metastatic Ovarian Carcinoma; Advanced/Metastatic Colorectal Carcinoma; Advanced/Metastatic Renal Cell Carcinoma; Advanced /Metastatic Pancreatic Carcinoma

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
