# Peer review of "Pathophysiology of Arginases in Cancer and Efforts in Their Pharmacological Inhibition"

_ijms, 2024, doi:10.3390/ijms25189782_

Round 1

Reviewer 1 Report

Comments and Suggestions for Authors

The manuscript titled "Molecular and Cellular Consequences of Arginases Inhibition in Cancer" presents an interesting review but could benefit from improved coherence in certain sections. While the title promises insights into the consequences of arginase inhibition, the text primarily focuses on the regulation and expression of arginases. Readers interested in the direct inhibition of Arg-1 and Arg-2 may find the current content misaligned with their expectations.

To address this, consider revising the title to more accurately reflect the content, such as: "Pathophysiology of Arginases in Cancer: Expression, Regulation, and the Future of Therapeutic Inhibitors."

Key Points for Improvement:

  • Main Topics Figure: Incorporate a figure summarizing the relationships between enzymes, substrates, and mediators (e.g., nuclear factors) to enhance understanding of the main topics.
  • Figure 1:
    • Group terms related to "Cell proliferation, Cell invasion, cell migration" on the right side, mirroring the left side's organization.
    • Improve resolution and remove any shadows from the base of the figure.
  • Figures 2 and 3: If Figure 3 is identical to Figure 2, remove it. Otherwise, improve its details to offer distinct information. For instance, clarify that OTC does not convert L-ornithine to polyamines. Consider including details about the polyamines pathway.
  • Topic 6.1 Arginase vaccines and pegylated rhARG in clinical trials: Clearly address the experimental procedures and potential adverse effects, such as the possibility of destroying constitutive cells (e.g., in the liver and kidneys) presenting endogenous arginase peptides via MHC-I.
  • Topics 6.2 and 6.3:
    • 6.1 Natural arginase inhibitors: Clarify that natural compounds are indeed chemicals.
    • 6.2 Chemical arginase inhibitors: Change the title to "Synthetic arginase inhibitors" for clarity.

Specific Text Revision:

  • Line 549: Change "oral arginase inhibitors" to "arginase inhibitors administered orally" to explicitly indicate the route of administration.

Author Response

Reviewer #1

The manuscript titled "Molecular and Cellular Consequences of Arginases Inhibition in Cancer" presents an interesting review but could benefit from improved coherence in certain sections. While the title promises insights into the consequences of arginase inhibition, the text primarily focuses on the regulation and expression of arginases. Readers interested in the direct inhibition of Arg-1 and Arg-2 may find the current content misaligned with their expectations.

To address this, consider revising the title to more accurately reflect the content, such as: "Pathophysiology of Arginases in Cancer: Expression, Regulation, and the Future of Therapeutic Inhibitors."

Response: Following the Reviewer suggestion the title of the manuscript was improved to reflect the content of the main text.

Line 1 – 3: “Pathophysiology of Arginases in Cancer and Efforts in their Pharmacological Inhibition”

Key Points for Improvement:

Main Topics Figure: Incorporate a figure summarizing the relationships between enzymes, substrates, and mediators (e.g., nuclear factors) to enhance understanding of the main topics.

Response: Following the Reviewer suggestion, in the revised version of the manuscript a new table (Table 1) summarizing the relationship between enzymes, substrates, mediator and processes was introduced.

Line 55: “ARG1 and ARG2 regulate essential processes involving several mediators”

Figure 1: Group terms related to "Cell proliferation, Cell invasion, cell migration" on the right side, mirroring the left side's organization. Improve resolution and remove any shadows from the base of the figure.

Response: Following the Reviewer suggestion, the shadows from the Figure 1 were removed and the terms were reorganized. Moreover, in the revised version of the manuscript, Figure 1 with a better resolution was uploaded.

Figures 2 and 3: If Figure 3 is identical to Figure 2, remove it. Otherwise, improve its details to offer distinct information. For instance, clarify that OTC does not convert L-ornithine to polyamines. Consider including details about the polyamines pathway.

Response: In fact, Figure 2 and Figure 3 are different from each other. In Figure 2, we presented how the urea cycle is proceed in cancers and what is the role of arginases in cancer metabolism. In contrast, Figure 3 described the effect of arginases inhibition on the urea cycle by a dual ARG1 and ARG2, cell permeable inhibitor (OATD-02) in cancers context. The above-mentioned changes were further highlighted using arrows and bold front. Moreover, according to the Reviewer suggestion additional sentence about ODC was added.

Line 604-605: “L-ornithine is not converted to polyamines by ODC, and by OAT to proline and glutamate”

Topic 6.1 Arginase vaccines and pegylated rhARG in clinical trials: Clearly address the experimental procedures and potential adverse effects, such as the possibility of destroying constitutive cells (e.g., in the liver and kidneys) presenting endogenous arginase peptides via MHC-I.

Response: Following the Reviewer suggestions several sentences about experimental procedures and adverse effects were added.

Line 465-467: “Vaccines were administered subcutaneously to patients every third week. The study used enzyme-linked immunosorbent assays and intracellular cytokine staining to assess vaccine-specific immune response.”

Line 470-474: “Adverse events reported in the study included injection site reactions and shoulder arthralgia [152-153]. There was a risk of inducing autoimmune reactions in patients, due to the fact, that ARG1 is expressed in the liver and other non-cancerous tissues. However, increased transaminase levels were noted only in patients with progressing metastatic lesions in the liver [152-153].”

Line 484-487: “PEG-BCT-100 was administered intravenously to patients at a dose of 2.7 mg/kg weekly [155]. Adverse events after administration of PEG-BCT-100 reported by patients included limb oedema, fatigue, constipation and anemia [155].”

Topics 6.2 and 6.3: 6.1 Natural arginase inhibitors: Clarify that natural compounds are indeed chemicals. 6.2 Chemical arginase inhibitors: Change the title to "Synthetic arginase inhibitors" for clarity.

Response: Following the Reviewer suggestion, the term “natural arginase inhibitors” was replaced by the term “naturally occurring arginase inhibitors”. Moreover, according to the Reviewer suggestion, the subtitle “chemical arginase inhibitors” was replaced by “synthetic arginase inhibitors”.

Line 495-496: “naturally occurring compounds”

Line 496: “Naturally occurring arginase inhibitors”

Specific Text Revision:

Line 549: Change "oral arginase inhibitors" to "arginase inhibitors administered orally" to explicitly indicate the route of administration.

Response: Following the Reviewer suggestion the term “oral arginase inhibitors” was replaced by the term “arginase inhibitors administered orally”.

Line 563-565: “Numidargistat (CB-1158) developed by Calithera Biosciences is a small molecule and arginase 1 inhibitor administrated orally with an IC50 of 98 nM [182-183].”

Reviewer 2 Report

Comments and Suggestions for Authors

                The authors have prepared a comprehensive review of arginases, covering multiple aspects. It is extensively referenced, and the figures are informative. It is somewhat long to wade through, but since it will probably be used as a starting point for readers interested in specific aspects of arginases, it will serve as a useful reference. I have only minor suggestions for improvement.

1) The quality of English is sufficiently good that the intended meaning is clear, for the most part, but will benefit from editing. However, on lines 56-57 there is a sentence fragment of uncertain meaning, suggesting that something has been omitted:

“Due to the that fact there is a growing interest in these enzymes in the area of drug discovery.”

The authors should correct this statement.

2) Knockout (KO) mice are a useful source of insight into gene function. PMID: 27761413 showed that the phenotype of mice with liver-specific KO of Arg1 is similar to that of mice with total-body KO of Arg1, supporting the authors’ claim that ARG1 is expressed mainly in the liver. This and other papers on KO of Arg1 and/or Arg2 appear not to have been cited. Some inclusion of these would improve the review.

Comments on the Quality of English Language

Nothing to add

Author Response

Reviewer #2

The authors have prepared a comprehensive review of arginases, covering multiple aspects. It is extensively referenced, and the figures are informative. It is somewhat long to wade through, but since it will probably be used as a starting point for readers interested in specific aspects of arginases, it will serve as a useful reference. I have only minor suggestions for improvement.

The quality of English is sufficiently good that the intended meaning is clear, for the most part, but will benefit from editing. However, on lines 56-57 there is a sentence fragment of uncertain meaning, suggesting that something has been omitted: “Due to the that fact there is a growing interest in these enzymes in the area of drug discovery.” The authors should correct this statement.

Response: Following the Reviewer suggestion the sentence was rephrased.

Line 59-61: “Accumulating evidence regarding the impact of arginases in many pathophysiological stages is linked with a growing interest in these enzymes in the area of drug discovery.”

Knockout (KO) mice are a useful source of insight into gene function. PMID: 27761413 showed that the phenotype of mice with liver-specific KO of Arg1 is similar to that of mice with total-body KO of Arg1, supporting the authors’ claim that ARG1 is expressed mainly in the liver. This and other papers on KO of Arg1 and/or Arg2 appear not to have been cited. Some inclusion of these would improve the review.

Response: Following the Reviewer suggestion data from experiment using KO mice were included in the revised version of the manuscript and appropriate references were added.

Line 446-448: “Additionally, it is worth noting that phenotypes of mice with liver-specific KO of Arg1 and phenotypes of mice with total-body KO of Arg1 are similar to each other and in both cases are lethal [146,147].”

Line 953-954: “Ballantyne L.L, Sin Y.Y, Al-Dirbashi O.Y, Li X, Hurlbut D.J, Funk C.D. Liver-specific knockout of arginase-1 leads to a profound phenotype similar to inducible whole body arginase-1 deficiency. Mol Genet Metab Rep 2016 Oct 12:9:54-60. ”

Line 955-956: “Ballantyne L.L, Sin Y.Y, Amand T.S, Si J, Goossens S, Haenebalcke L, Haigh J.J, Kyriakopoulou L, Schulze A, Funk C.D Strategies to rescue the consequences of inducible arginase-1 deficiency in mice. PLoS One 2015 May 4;10(5):e0125967. ”

Round 2

Reviewer 1 Report

Comments and Suggestions for Authors

1) Please, consider revision on biochemistry technical language:

“L-ornithine is not converted to polyamines by ODC in Line 604-605: “L-ornithine is not converted to polyamines by ODC, and by OAT to proline and glutamate”

Example:

The enzyme ornithine decarboxylase (EC 4.1.1.17, ODC) catalyzes the decarboxylation of ornithine (a product of the urea cycle) to form putrescine.

2) Line 563-565: “Numidargistat (CB-1158) developed by Calithera Biosciences is a small molecule and arginase 1 inhibitor, administrated orally, with an IC50 of 98 nM [182-183].”

Suggestion for revision:

administrated orally, or (administrated orally)

Comments on the Quality of English Language

The quality of the manuscript was improved, but Moderate editing of the English language is required.

Author Response

  • Please, consider revision on biochemistry technical language:

"L-ornithine is not converted to polyamines by ODC in Line 604-605: "L-ornithine is not converted to polyamines by ODC, and by OAT to proline and glutamate”

Example:

The enzyme ornithine decarboxylase (EC 4.1.1.17, ODC) catalyzes the decarboxylation of ornithine (a product of the urea cycle) to form putrescine.

Response: According to the Reviewer suggestion the sentence " Inhibition of both arginases causes reduced production of L-ornithine (L-ornithine is not converted to polyamines by ODC, and by OAT to proline and glutamate), which translates into reduced polyamines, proline and glutamate generation, thus inhibiting further proliferation of cancer cells” was changed.

Line 607-611: "Inhibition of both arginases causes reduced production of L-ornithine. ODC does not catalyze the decarboxylation of L-ornithine to form putrescine, a substrate necessary for the synthesis of polyamines, which is translates into reduced polyamines generation, thus inhibiting further proliferation of caner cells. Additionally, OAT does not catalyze the reaction of converting L-ornithine to 5PC, which translates into reduced proline and glutamate generation.”

EC numbers for enzymes were described in the main text and Figure 2.

Line 79: "ornithine decarboxylase (ODC, 4.1.1.17)”

Line 82: "arginine decarboxylase (ADC, 4.1.1.4)”

Line 103: "ornithine aminotransferase (OAT, 2.6.1.13)”

Line 119: "nitric oxide synthase (NOS, EC 1.14.13.39)”

Line 226: "argininosuccinate synthase (ASS, EC 6.3.4.5)”

Line 226: "ornithine transcarbamylase (OTC, EC 2.1.3.3)”

Line 396: "glycine amidinotransferase AGAT (EC 2.1.4.1)”

Line 397: "guanidinoacetate methyltransferase GAMT (EC 2.1.1.2)”

Line 407: "ASL (EC 4.3.2.1), argininosuccinate lyase”

Line 409: "P5Cr (EC 1.5.1.2), P5C reductase”

Line 409: "P5CS (EC 2.7.2.11), P5C synthase”

  • Line 563-565: “Numidargistat (CB-1158) developed by Calithera Biosciences is a small molecule and arginase 1 inhibitor, administrated orally, with an IC50 of 98 nM [182-183].”

Suggestion for revision:

,administrated orally, or (administrated orally)

Response: Following the Reviewer suggestion the sentence was rephrased.

Line 566-568: “Numidargistat (CB-1158) developed by Calithera Biosciences is a small molecule and arginase 1 inhibitor, administrated orally, with an IC50 of 98 nM [182-183].”